# A Dedicated Expert ECMO-Team and Strict Patient Selection Improve Survival of Patients with Severe SARS-CoV-2 ARDS Supported by VV-ECMO

**DOI:** 10.3390/jcm12010230

**Published:** 2022-12-28

**Authors:** Raphaël Giraud, Hannah Wozniak, Viviane Donner, Carole Looyens, Benjamin Assouline, Karim Bendjelid

**Affiliations:** 1Intensive Care Unit, Geneva University Hospitals, CH-1205 Geneva, Switzerland; 2Geneva Hemodynamic Research Group, Faculty of Medicine, University of Geneva, CH-1211 Geneva, Switzerland; 3Surgical Intensive Care Division, Faculty of Medicine, University of Geneva, CH-1211 Geneva, Switzerland

**Keywords:** ARDS, VV-ECMO, SARS-CoV-2 pneumonia, COVID-19 pandemic

## Abstract

The SARS-CoV-2 pandemic has overwhelmed health care systems worldwide since its first wave. Intensive care units have been under a significant amount of pressure as patients with the most severe form of the disease presented with acute respiratory distress syndrome (ARDS). A proportion of them experienced refractory acute respiratory failure and had to be supported with veno-venous extracorporeal membrane oxygenation (VV-ECMO). The present retrospective study reports the experiences of our ECMO center in the management of COVID-19 patients with refractory ARDS. Patient characteristics and outcomes are presented through the different waves of the pandemic. A cohort study was conducted on patients with refractory ARDS due to COVID-19 infection who were admitted to the intensive care unit (ICU) at the Geneva University Hospital and supported with VV-ECMO between 14 March 2020 and January 2022. The VV-ECMO implementation criteria were defined according to an institutional algorithm validated by the local crisis unit of the hospital and the Swiss Society of Intensive Care Medicine. Among the 500 ARDS patients admitted to our ICU, 41 patients with a median age of 57 (52–63) years, a body mass index (BMI) of 28 (26–32) kg/m^2^, and a SAPS II score of 57 (47–67), and 27 (66%) of whom were men required VV-ECMO. None of the patients were vaccinated. The time of ventilation, including noninvasive ventilation (NIV) and mechanical ventilation (MV), and the time of MV before ECMO were 7 (4–11) days and 4 (1–7) days, respectively. The time under ECMO was 20 (10–27) days. The ICU and hospital lengths of stay were 36 (21–45) days and 45 (33–69) days, respectively. The survival rate for patients on ECMO was 59%. Comparative analysis between survivors and non-survivors highlighted that survivors had a significantly shorter ventilation duration before ECMO (NIV + MV: 5.5 (1.3–9) vs. 9 (6.5–13.5) days, *p* = 0.0026 and MV alone: 1.6 (0.4–5.5) vs. 5.8 (5–8) days, *p* < 0.0001). The management of patients on ECMO by an experienced ECMO team dedicated to this activity was associated with improved survival (78% vs. 28%, *p* = 0.0012). Between the first wave and the following waves, patients presented with a higher incidence of ventilator-associated pneumonia (100% vs. 82%, *p* = 0.0325) but had better survival rates (74% vs. 35%, *p* = 0.024). The present study suggests that both the prompt insertion of VV-ECMO to control refractory hypoxemia and the involvement of an ECMO team improve the survival of COVID-19 patients.

## 1. Introduction

The SARS-CoV-2 pandemic has been ongoing for two years and has kept intensive care units (ICUs) quite busy with patients with the most severe form of the disease presenting with ARDS due to COVID-19 pneumonia [1]. The massive influx of patients has forced ICUs to significantly increase their number of beds and, therefore, to constantly reorganize over the different waves to be able to treat all affected patients [2]. Among these patients, some presented refractory forms of hypoxemia and had to be supported with veno-venous extracorporeal membrane oxygenation (VV-ECMO) [3]. This extracorporeal assistance initially showed mixed results in terms of survival [4], but several retrospective studies have now shown its benefit for assisting patients who do not respond to optimal medical care [5,6].

In Geneva, more than 500 COVID-19 patients have been admitted to ICUs. Initially, the ICU was staffed with 30 beds divided into 4 specialized units; however, it accepted up to nearly 70 intubated patients during the first wave, including 10 under VV-ECMO [3,7]. The internal organization of the ICU, the supervision of patients on ECMO, and the criteria for implantation have varied during the different waves depending on the number of COVID-19 patients, the planned surgical activity, and the results of published studies.

The present retrospective study reports the ECMO activity of our ICU and its evolution regarding COVID-19 patients with severe ARDS, their characteristics, patient outcomes, and differences that may have occurred during different waves of the pandemic.

## 2. Materials and Methods

### 2.1. Study Design

Our ICU normally holds 30 beds divided into 4 functional sectors, including one that specializes in intensive cardiovascular medicine which supports approximately 50 patients per year with ECMO. In addition, a mobile ECMO unit provides care, support, and retrieval for several regional hospitals in Switzerland. Forty-one patients were included in the analysis who had a confirmed SARS-CoV-2 infection based on a positive nasopharyngeal and/or endobronchial PCR test and were supported with ECMO (veno-venous or veno-arterial ECMO) in the context of refractory ARDS or cardiogenic shock (e.g., myocarditis or pulmonary embolism). Data from 10 patients treated in our ICU which was previously reported in a retrospective cohort study [3] were also included in this study. Medical management evolved during the pandemic according to international guidelines, including specific treatments for COVID-19. Hydroxychloroquine and the lopinavir/ritonavir combination used systematically in our department during the first wave were initially replaced with remdesivir and then with dexamethasone and tociluzimab during the second wave. When indicated, the Meduri protocol [8] was used.

The Cantonal Ethics Commission for Human Research approved this study and waived the requirement for informed consent (BASEC number: 2020-00917).

### 2.2. ECMO Organization

Switzerland is a federal country. Responsibilities for managing the health system are shared between the Swiss Confederation and cantonal authorities. The Confederation sets the regulatory framework and notably manages compulsory health insurance and the fight against communicable diseases. Cantons apply the federal regulations but have vast competences in the health field, particularly regarding hospital care, advanced medicine, the exercise of health professions, and disease prevention. Each canton had to organize itself to address a significant influx of patients during the first wave. However, the burden differed significantly between cantons, and some were more quickly and severely impacted by the pandemic. A national coordination in intensive care medicine was established quickly to identify the number of ICU beds across the country; however, it was only during the second wave that this coordination unit was commissioned to help transfer patients from overburdened centers to other centers with more available beds.

Regarding ECMO activity, guidelines were edited and nationally distributed under the guidance of the Swiss Society of Intensive Medicine. A formal network with non-ECMO centers was therefore established. Practices were standardized which allowed centers to nationally organize cannulations and transfers of critical patients to ECMO centers.

In our center, the ECMO team included two intensivist consultants who are experts in the cannulation and management of ECMO patients. They set up 24/7 ECMO on-call duties and trained a younger physician in this expertise during the first wave, which enabled him to join this ECMO team during subsequent waves. In addition, the team included 10 nurses who were specifically trained in the care of ECMO patients.

The ECMO team was not able to ensure the direct care of all patients with ECMO during the first wave due to its specific organization. Indeed, ICU capacity drastically increased to over 100 beds and was divided in an independent unit. Afterward, during the following waves, the lower number of COVID-19 patients allowed the ECMO team to take care of the majority of ECMO patients in the cardiovascular ICU.

### 2.3. ECMO Indications for COVID-19

Eligibility criteria for VV-ECMO for COVID-19 ARDS applied in Switzerland were published in a previous article [3]. It is worth noting that the maximum duration of mechanical ventilation (MV) before ECMO implantation was set at 10 days during the first wave. This duration was reduced to 7 days during and following the second wave [3].

### 2.4. Cannulation and Bed Management

The ECMO consultant was contacted when patients hospitalized in our institution or in a non-ECMO center fulfilled the ECMO eligibility criteria. The consultant had to validate the indication and consider ICU bed availability, human resources (i.e., experienced ICU nurses available 24/7), local resources (e.g., power supply, medical gases, and space), technical resources (e.g., pump, cannula, and circuit availability), and Swiss Air Rescue (REGA) availability for retrieval. A mobile ECMO team, including an intensivist from the ECMO team and a perfusionist, was sent to patients in non-ECMO centers to perform echo-assisted cannulation and then transport patients to our hospital.

### 2.5. Management of ECMO Patients

Recommendations for MV settings during ECMO support were to practice ultra-protective ventilation with a reduction of Vt < 4 mL/kg/PBW and a level of PEEP > 10 cmH_2_O in order to maintain a driving pressure < 12 cmH_2_O, a respiratory rate < 12/min, and a FiO_2_ < 40% by adjusting the ECMO flow. The use of transpulmonary pressure monitoring was recommended during and following the second wave.

Anticoagulation levels were adjusted according to the patients’ condition and monitored by daily measurement of antiXa activity, including 0.2–0.3 IU/mL for patients without thromboembolic disease (TED) and 0.3–0.4 IU/mL for patients with TED. Temporary discontinuation of anticoagulation was recommended in cases of bleeding. The transfusion threshold was set at 7 g/dL of hemoglobin. The transfusion of platelet concentrates was not recommended except in cases of thrombocytopenia accompanied by bleeding that required a red cell transfusion. Postmembrane gas exchange was monitored daily by the perfusionists. Free hemoglobin and fibrinogen levels were checked regularly, and the carbon monoxide level [9] was checked regularly to assess membrane wear. A postmembrane exit PaO_2_ < 30 kPa and a decrease in fibrinogen < 1.5 g/L associated with thrombocytopenia with a decrease of more than 30% in the platelet count were arguments in favor of a change of circuit by the intensivists on the ECMO team.

### 2.6. Data Collection and Outcomes

The data were extracted from institutional electronic medical records, the Patient Data Management System (PDMS), and Centricity critical care (GE Healthcare, General Electric Company, Boston, MA, USA). The collected data included baseline patient characteristics; risk factors; comorbidities before COVID-19; the SAPS II score on ICU admission; the specific COVID-19 treatment; hemodynamic, respiratory, and biological parameters on admission and before ECMO setup, including treatment with rescue therapies and ventilator settings; and indication for continuous renal replacement therapy (*CRRT*). The durations between initial symptoms and hospitalization (in the ward vs. intermediate care vs. ICU) were also extracted, as was the use of the mobile ECMO team, the parameters and duration of ECMO support, specific COVID-19- and ECMO-related complications, ICU length of stay and hospital length of stay, survival status at 90 days after initiation of ECMO, causes of death, and outcomes.

### 2.7. Statistical Analysis

We performed descriptive analyses of patients’ characteristics according to 90-day survival status and the wave of hospitalization (i.e., whether it was the first wave or the following waves). Continuous variables are presented as the median and the interquartile range (IQR), and categorical variables are presented as the number of patients (n) and the percentage (%). The *t* test for normally distributed variables and the Mann-Whitney *U* test for non-normally distributed variables were used for the continuous variables. The chi-square test or the Fisher’s exact test was used to detect differences in categorical variables. 

We performed a multivariate logistic regression and adjusted the estimates for SAPSII and Pplat on ECMO implantation to investigate the association between time from intubation to ECMO. Variables entered into the multivariable model were defined a priori as possible confounding factors, regardless of their univariate p value, on the basis of published literature on ECMO and COVID-19. The results are expressed as odds ratios (ORs) and 95% confidence intervals (CIs). Two-tailed *p* values < 0.05 were considered statistically significant.

Finally, Kaplan-Meier plots were performed for variables that were significantly associated with survival and for those that seemed relevant based on the literature (e.g., the duration of MV before ECMO implantation, first wave vs. other waves, and whether the ECMO team was or was not in charge). All patients were censored at 90 days after ECMO. Univariable analysis of the factors associated with 90-day overall survival was performed using log-rank tests. Two-tailed *p* values < 0.05 were considered statistically significant. All analyses were performed using GraphPad Prism 6 for Windows (GraphPad Software, San Diego, CA, USA), and logistic regression was conducted using STATA version 16.1 (Stata Corp., College Station, TX, USA, 2007).

## 3. Results

Forty-one COVID-19 patients required ECMO in our ICU between March 2020 and January 2022. Thirty-nine required VV-ECMO in the setting of refractory COVID-19 associated-ARDS, and two required VA-ECMO in the context of refractory cardiogenic shock (including one myocarditis and one massive pulmonary embolism with thrombolysis failure). The median age (IQR) of the patients placed on ECMO was 57 (52–63) years. Their BMI was 28 (26–32) kg/m^2^, and their SAPS II score on admission was 57 (47–67). Thirteen (34%) patients were treated chronically for high blood pressure, while 14 (37%) were treated for diabetes, 14 (37%) were obese, and 10 (26%) had no comorbidities. The time from symptom presentation to hospital admission, the time from hospital to intermediate care (IMC), and the time from IMC to ICU were 5 (3–7) days, 0 (0–1) days, and 2 (0–4.5) days, respectively. Clinical characteristics, comorbidities, laboratory parameters, pre-ECMO and on-ECMO parameters, and outcomes are presented in Table 1. Pre-ECMO patients were ventilated using a controlled volume with protective ventilation (Vt 6.2 (5.7–6.7) mL of PBW) and an optimized PEEP level (12 (10–12) cmH_2_O) under neuromuscular blockers for the first 48 h. Patients had low compliance (24 (20–27) ml/cmH_2_O) and were severely hypoxemic with a P/F ratio of 9.5 (7.7–11.1) kPa. The Murray score was 3.5 (3.5–3.8), and the percentage of lung lesions on CT scans was 70% (60–75%). The number of prone positioning sessions per patient was 1 (1–2.5) before ECMO implantation.

The total ventilation time (including NIV and MV) and MV time alone before ECMO were 7 (4–11) days and 4 (1–7) days, respectively, while the median time under ECMO was 20 (10–37) days, and the ICU and hospital lengths of stay were 36 (21–45) days and 45 (33–69) days, respectively. The 90-day survival rate for patients put on ECMO was 59%. Only two major adverse events occurred: one patient developed a massive cerebral hemorrhage without neurosurgical sanction, and the second patient experienced accidental decannulation during mobilization, which both led to death. The other deaths occurred after therapeutic withdrawal in the context of extensive pulmonary fibrosis or refractory septic shock.

The characteristics and differences between survivors and non-survivors are presented in Table 1. Survivors had significantly lower D-dimer values (1039 (726–1924) vs. 4302 [1925–9681] ng/mL, *p* = 0.01); shorter ventilation (MV + NIV) and MV times before ECMO (5.5 [1.3–9] vs. 9 [6.5–13.5) days, *p* < 0.01 and 1.6 (0.4–5.5) vs. 5.8 (5–8) days, *p* < 0.01, respectively); lower plateau pressures (28 (24–30) vs. 30 (28–32) cmH_2_O, *p* = 0.01); higher compliance (25 (21–32) vs. 22 (17–25) ml/cmH_2_O); and fewer prone sessions (1 (0–1) vs. 3 (1–4), *p* < 0.01) before ECMO implantation. On ECMO, survivors had significantly fewer prone sessions (1 (0–1) vs. 3 (1–4), *p* < 0.01), were more closely followed and monitored by the ECMO team (18 (75%) vs. 5 (21%)), and had lower sweep gas flows on the membrane (8 (7–10) vs. 9 (8–14) L/min, *p* = 0.041). Regarding COVID-19-related complications, survivors had a significantly lower incidence of pneumothorax (5 (21%) vs. 9 (53%)) than non-survivors (*p* = 0.03).

The results of the multivariate logistic regression assessing the association between time under MV before ECMO and mortality are presented in Table 2. Being under MV for more than 3 days before ECMO treatment was significantly associated with mortality after adjusting for the SAPSII score and Pplat at ECMO implantation (as shown in Figure 1 and Figure 2).

Comparative analyses between the first wave and the following waves highlighted that patients had significantly lower CRP and procalcitonin levels after the first wave. Regarding MV settings, Vt, Vt/PBW, Pplat, and compliance were significantly higher. Patients after the first wave had both shorter ventilation (MV + NIV) and MV times before ECMO (6 (2–9) vs. 9 (6.5–13.59 days, *p* = 0.0249 and 1.5 (0–4.8) vs. 7 (5–8.5) days, *p* < 0.01, respectively). Patients were ventilated with significantly lower Pplat and respiratory rates on ECMO. Patients were more often managed by the ECMO team (66% vs. 34%, *p* = 0.01). Survival rates differed significantly depending on the expertise of the team in charge of ECMO patients (ECMO team vs. non-ECMO team, 78% vs. 28%, *p* = 0.0012) (Table 3 and Figure 3). Finally, there were significantly higher incidences of VAP (100% vs. 82%, *p* = 0.0325) and pulmonary embolism (10% vs. 0%, *p* < 0.01) after the first wave. Nonetheless, the survival rate was higher (74% vs. 35%, *p* = 0.02).

## 4. Discussion

The present single-center, retrospective cohort study included 41 patients with COVID-19 associated-ARDS supported with VV-ECMO. It showed an overall 90-day survival rate of 59% (24/41) and highlighted that both the time between intubation and ECMO implantation ≤72 h and the management of these patients by a dedicated experienced ECMO team were significantly associated with survival.

The 90-day survival of our cohort is similar to what was reported in the literature in high-volume centers [5,6,10,11,12]. 

ECMO centers in Switzerland agreed on the implantation criteria and contraindications to the implementation of ECMO in COVID-19 patients at the start of the pandemic. These criteria were applied in our center. However, these recommendations did not specify the maximum duration of MV as no data existed at that time regarding COVID-19 associated-ARDS. We limited the implantation of ECMO to patients who had MV for less than 7 days following the first wave and the first published data and based on our experience. Since then, several published studies have shown that the sooner ECMO is implanted after intubation and MV, the better the survival. Our results also support an early implantation of ECMO with results showing increased mortality after more than 3 days of MV before ECMO implantation. This study also underlines that the prognosis for survival is improved with early implantation of ECMO, even if hypoxemia occurs later. Indeed, the early implantation of VV-ECMO in severe ARDS allows faster initiation of ultraprotective ventilation which prevents mechanical damage to the lung parenchyma due to artificial ventilation while improving oxygenation, redox stress, and decarboxylation.

Our results show that the total duration of ventilation (NIV + MV) also has an impact on survival. The use of NIV in hypoxemic respiratory failure due to COVID-19 pneumonia is a debated topic. Central to this debate are the role of NIV in preventing intubation in patients with mild respiratory disease and its potential beneficial effects on both patient outcome and resource utilization. However, there remains valid concern that the use of NIV may prolong time to intubation and its associated lung protective ventilation in patients with more advanced disease, thereby worsening respiratory mechanics via self-inflicted lung injury [13]. In addition, the risk of self-inflicting lung injury (SILI) and aerosolization with the use of NIV has the potential to increase health care worker (HCW) exposure to the virus. Strict monitoring of these measures seems essential to limit their deleterious effects and their negative impact on survival.

This study also shows that mortality is associated with the experience of the team in charge of patients placed on ECMO. The positive effect of ECMO case volume on outcomes has been reported previously [14,15,16] and is emphasized in this study due to the fact that all the indications were validated by the same expert ECMO team using strict and standardized criteria. In a multicenter cohort French study, the authors observed that mortality due to COVID-19 associated ARDS varied across centers, despite a strategy of central regulation and the use of identical VV-ECMO protocol. Interestingly, they showed that survival was not dependent of the total number of ECMO procedures performed each year (VA and VV-ECMO) per center; rather, it was dependent on the specific volume of VV-ECMO cases [6]. These results show that VV-ECMO requires specific expertise that is not naturally obtained in cardiovascular centers with mainly VA-ECMO experience. In light of these findings, every possible effort must be made in the organization of an ICU to allow ECMO patients to be cared for by an experienced team dedicated to this specific activity.

However, it is important to point out that an ECMO case threshold that defines a high-volume ECMO center is still unknown. A retrospective analysis of an international ECMO registry reported that patients receiving ECMO in hospitals with more than 30 ECMO cases per year had a significantly lower mortality risk than adults receiving ECMO in hospitals with less than six cases per year [14]. Conversely, as published by an expert Parisian team, we observed no significant difference in survival between cannulated patients transferred to an ECMO center by our mobile teams and patients cannulated and supported on site, which validates the concept and effectiveness of mobile ECMO teams [17].

Our study has some limitations. First, our analyses are retrospective and limited to the recorded data. Nevertheless, there are no missing data among the collected variables. Second, we cannot externally validate our results as this was a single-center study. Third, the results of statistical analyses should be interpreted with caution due to the study’s relatively small sample size. Larger cohort studies should make it possible to clarify certain important aspects, such as the timing of implementation of ECMO during the evolution of respiratory illness in these patients. Despite these limitations, this study is, to our knowledge, the first to show that a duration of pre-ECMO MV greater than 150 h is associated with mortality, even after adjusting for possible confounding factors.

## 5. Conclusions

The present study suggests that early implantation of VV-ECMO in highly selected patients with refractory hypoxemia related to SARS-CoV-2-ARDS and management of these patients by a dedicated team experienced in ECMO significantly improves survival.

## Figures and Tables

**Figure 1 jcm-12-00230-f001:**
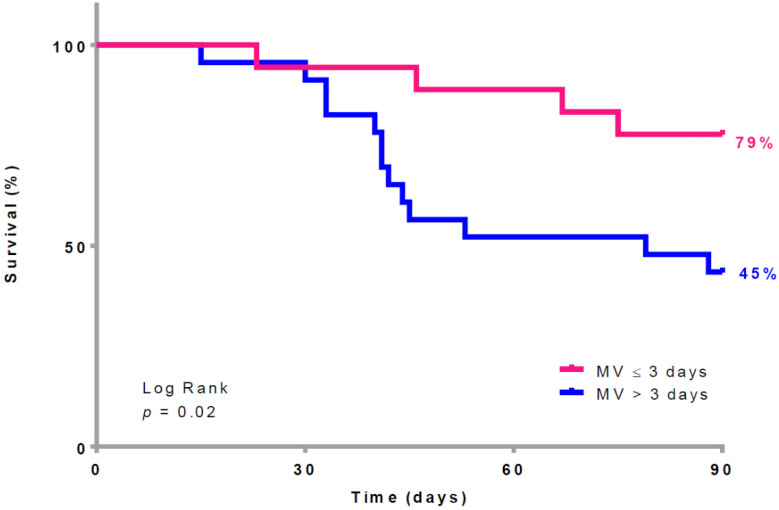
90-day survival according to the time between mechanical ventilation and ECMO implantation. ECMO: extracorporeal membrane oxygenation; MV: mechanical ventilation.

**Figure 2 jcm-12-00230-f002:**
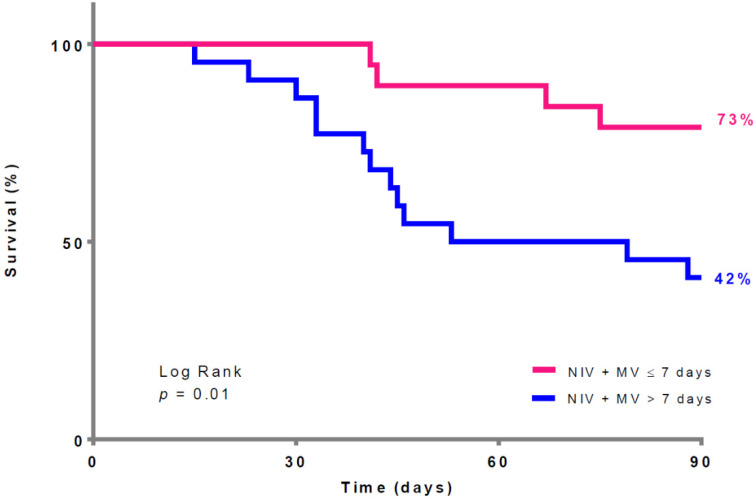
90-day survival according to the time between non-invasive ventilation + mechanical ventilation and ECMO implantation. ECMO: extracorporeal membrane oxygenation; NIV: non-invasive ventilation; MV: mechanical ventilation.

**Figure 3 jcm-12-00230-f003:**
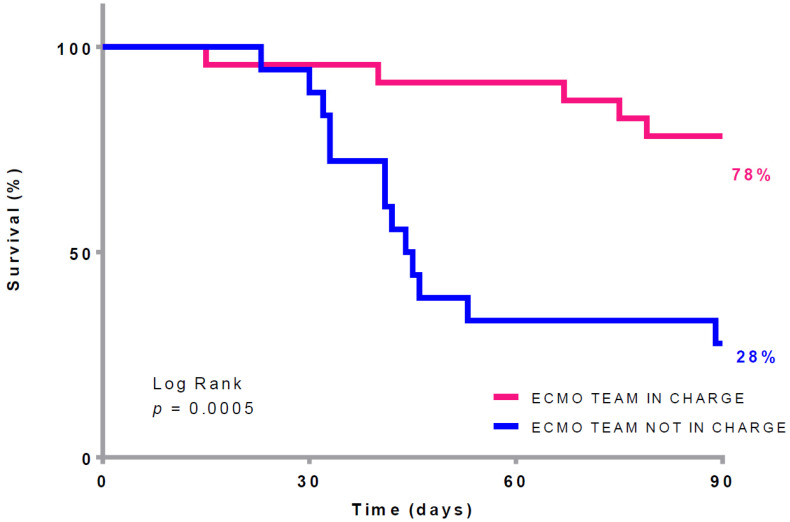
90-day survival according ECMO patients managed or not managed by an experienced and dedicated team. ECMO: extracorporeal membrane oxygenation.

**Table 1 jcm-12-00230-t001:** Overall demographic, clinical, biological, hemodynamic, and mechanical ventilation characteristics before and on ECMO according to 90-day survival status and depending on the wave vs. other waves).

Clinical Characteristics	All (*n* = 41)	Survivors (24)	Non-Survivors (17)	*p*	First Wave (17)	Other Waves (24)	*p*
At ICU admission							
Gender (female/male)	14/27	10/14	4/13	0.01	7/10	7/17	NS
Age (years)	57 (52–63)	57 (47–61)	58 (53–66)	NS	**57 (54–64)**	57 (47–61)	NS
-BMI (kg/m^2^)	28 (26–32)	28.5 (26–31.75)	28 (25–34.5)	NS	26.2 (24.9–29.4)	29.4 (26.6–32.5)	NS
-SAPS II	57 (47–67)	56 (38–63)	59 (52–67)	NS	**59 (53–68)**	**56 (38–63)**	**0.0317**
Comorbidities							
Hypertension n (%)	13 (34)	9 (38)	4 (24)	NS	6 (35)	7 (29)	NS
Diabetes n (%)	14 (37)	8 (33)	6 (35)	NS	5 (29)	9 (38)	NS
Obesity n (%)	15 (37)	10 (42)	5 (29)	NS	**3 (18)**	**12 (50)**	**0.0182**
Immunosuppression n (%)	4 (11)	2 (8)	2 (12)	NS	**4 (24)**	**0 (0)**	**0.0124**
Asthma/COPD n (%)	2 (5)	1 (4)	1 (6)	NS	2 (12)	0 (0)	NS
None n (%)	10 (26)	5 (21)	5 (29)	NS	2 (12)	8 (33)	NS
Time from symptoms to hospital admission (days)	5 (3–7)	5 (3–7)	6 (3.5–7.5)	NS	**4 (3–7)**	**6 (5–9)**	**0.0279**
Time from hospital to IMC (Days)	0 (0–1)	0 (0–1)	0 (0–1)	NS	0 (0–0.5)	0 (0–1)	NS
Time from IMC to ICU (Days)	2 (0–4.5)	2 (0–3)	3 (1–7)	NS	1 (0–5)	3 (1–5)	NS
Laboratory							
Hb (g/L)	126 (98–139)	129 (109–139)	102 (94–132)	NS	**96 (87–105)**	**131 (125–146)**	**<0.0001**
WCC (G/L)	9.5 (6.1–11.7)	8.5 (5.5–10)	10.6 (9.2–12.9)	NS	9.6 (5.3–12.7)	9.4 (6.9–10.5)	NS
Lymphocytes (G/L)	0.59 (0.28–0.81)	0.5 (0.3–0.7)	0.7 (0.6–0.9)	NS	0.56 (0.2–0.85)	0.62 (0.28–0.75)	NS
N/L ratio	17 (9–25)	19 (9–28)	17 (10–21)	NS	17 (9–41)	17 (10–23)	NS
Thrombocytes (G/L)	275 (192–347)	265 (188–324)	276 (234–374)	NS	275 (183–354)	278 (194–348)	NS
Ddimers (ng/mL)	2421 (972–8090)	**1039 (726–1924)**	**4302 (1925–9681)**	**0.01**	3411 (1775–8769)	1039 (723–4625)	NS
CRP (mg/L)	95 (38–208)	93 (42–173)	168 (25–250)	NS	**171 (65–306)**	**64 (23–130)**	**0.003**
Procalcitonin (µg/L)	0.22 (0.15–0.72)	0.2 (0.14–0.78)	0.25 (0.17–0.63)	NS	**0.29 (0.21–1.08)**	**0.18 (0.08–0.44)**	**0.0055**
Us-troponin (ng/L)	23 (7–49)	23 (6–42)	25 (15–107)	NS	28 (17–57)	13 (5.8–41)	NS
Creatinine (µmol/L)	63 (55–76)	64 (57–77)	61 (54–77)	NS	62 (54–79)	63 (57–74)	NS
Urea (mmol/L)	7.9 (5.6–9.2)	7.1 (5.2–8.3)	8 (7.2–10.5)	NS	8.1 (6.5–10.5)	7.4 (5.2–8.1)	NS
Lactate (mmol/L)	1.4 (1.2–2)	2 (0–3)	3 (1–7)	NS	1.5 (1.2–2.1)	1.4 (1.1–1.8)	NS
At ECMO implantation							
HR (BPM)	89 (79–100)	87 (78–99)	94 (79–101)	NS	81 (78–99)	91 (80–102)	NS
MAP (mmHg)	75 (72–77)	75 (72–77)	74 (68–78)	NS	75 (70–18)	75 (72–77)	NS
Norepinephrine (µg/kg/min)	0.08 (0.04–0.12)	0.08 (0.03–0.12)	0.09 (0.03–0.12)	NS	0.08 (0.01–0.11)	0.1 (0.04–0.12)	NS
Temperature (°C)	36.9 (36.6–37.1)	36.9 (36.6–37)	36.9 (36.5–37.5)	NS	37 (36.6–27.8)	36.8 (36.5–37)	NS
Lung lesions on CT-Scan (%)	70 (60–75)	75 (53–75)	75 (75–75)	NS	75 (68–75)	75 (53–75)	NS
Ventilatory mode, nb (%)	ACV 41 (100)	ACV 24 (100)	ACV 17 (100)	NS	ACV 17 (100)	ACV 24 (100)	NS
P/F ratio before ECMO (kPa)	9.5 (7.7–11.1)	10 (8–12)	8.3 (7.2–10.1)	NS	10 (8–11.5)	9 (7.3–11.1)	NS
Vt (mL)	380 (343–465)	405 (346–478)	370 (335–423)	NS	**370 (340–388)**	**450 (352–480)**	**0.0132**
Vt/PBW (mL/kg)	6.2 (5.7–6.7)	6.4 (5.7–7.1)	6.2 (4.9–6.6)	NS	**5.8 (4.7–6.5)**	**6.4 (6.2–7.1)**	**0.031**
Plateau pressure (cmH_2_O)	29 (27–31)	**28 (24–30)**	**30 (28–32)**	**0.01**	**29 (28–31)**	**28 (24–31)**	**0.045**
PEEP (cmH_2_O)	12 (10–12)	11 (10–12)	12 (10–13)	NS	12 (10–12)	11 (10–13.5)	NS
Respiratory rate (Breath/min)	23 (20–28)	23 (20–27)	24 (20–28)	NS	25 (20–29)	22 (20–26)	NS
Compliance (mL/cmH_2_O)	24 (20–27)	**25 (21–32)**	**22 (17–25)**	**0.03**	**21 (17–24)**	**25 (22–33)**	**0.001**
Prone positioning/patient	1 (1-2.5)	**1 (0-1)**	**3 (1-4)**	**0.0045**	8 (47)	13 (54)	NS
iNO (nb)	23 (56)	11 (46)	12 (72)	NS	12 (71)	11 (46)	NS
Neuromuscular blockers (nb)	40 (98)	23 (96)	17 (100)	NS	17 (100)	23 (96)	NS
Mechanical ventilation ≤3 days, nb (%)	**19 (46.3)**	**15 (62.5)**	**4 (23.5)**	**0.025**	**2 (11.8)**	**17 (70.8)**	**<0.0001**
Mechanical ventilation >3 days, nb (%)	**22 (53.6)**	**9 (37.5)**	**13 (76.5)**	**0.025**	**15 (88.2)**	**7 (29.2)**	**<0.0001**
Total ventilation time before ECMO (days)	7 (4–11)	**5.5 (1.3–9)**	**9 (6.5–13.5)**	**0.0026**	**9 (6–13.5)**	**6 (2–9)**	**0.0249**
Duration of mechanical ventilation before ECMO (days)	4 (1–7)	**1.6 (0.4–5.5)**	**5.8 (5–8)**	**<0.0001**	**7 (5–8.5)**	**1.5 (0–4.8)**	**<0.0001**
On ECMO							
Ventilatory mode, nb (%)	ACV 41 (100)	ACV 24 (100)	ACV 17 (100)	NS	ACV 17 (100)	ACV 24 (100)	NS
Vt (mL)	320 (210–345)	320 (225–355)	300 (210–340)	NS	320 (265–360)	320 (200–345)	NS
Vt/PBW (mL/kg)	4.3 (3.8–5.6)	4.7 (3.9–5.7)	4.3 (3.6–4.8)	NS	4.7 (4.2–5.7)	4.3 (3.5–5.3)	NS
Plateau pressure (cmH_2_O)	24 (22–26)	24 (20.3–25.8)	24 (22.5–26.5)	NS	**25 (23.5–29.5)**	**24 (20–24)**	**0.0126**
PEEP (cmH_2_O)	10 (10–12)	10 (10–12)	10 (10–13)	NS	10 (10–11.5)	10 (8.5–12)	NS
Respiratory rate (Breath/min)	11 (10–19)	11 (10–17)	10 (10–20)	NS	**18 (11–24)**	**10 (10–12)**	**0.002**
Compliance (mL/cmH_2_O)	23 (17–30)	23 (18–31)	22 (15–29)	NS	22 (15–29)	25 (18–31)	NS
Prone positioning, nb (%)	**21 (51)**	**1 (0–1)**	**3 (1–4)**	**0.0021**	8 (47)	13 (54)	NS
ECMO team in charge, nb (%)	**23 (66)**	**18 (75)**	**5 (21)**	**0.0012**	**5 (29)**	**19 (79)**	**0.0009**
Mobile ECMO, nb (%)	12 (29)	7 (29)	5 (29)	NS	5 (29)	7 (29)	NS
Reinjection cannula (French)	19 (19–21)	20 (19–21)	19 (19–21)	NS	**19 (19–19)**	**21 (19–21)**	**0.0068**
Drainage cannula (French)	28 (25–29)	27.5 (25–29)	26.5 (25–29)	NS	**25 (25–28)**	**29 (25–29)**	**0.0041**
AVALON cannula, nb (%)	9 (22)	6 (25)	3 (18)	NS	**1 (6)**	**8 (33)**	**0.0364**
VA-ECMO, nb (%)	2 (5)	2 (5)	0 (0)	NS	0 (0)	2 (5)	NS
VV-ECMO, nb (%)	39 (95)	22 (95)	17 (100)	NS	17 (100)	22 (95)	NS
ECMO blood flow (L/min)	5.0 (4.5–5.9)	5 (4.1–5.6)	5.5 (4.8–6)	NS	5 (4.8–5.6)	5.1 (4–6)	NS
Sweep gas flow (mL/min)	**8 (7–10)**	**8 (7–10)**	**9 (8–14)**	**0.041**	8 (7.5–10)	8 (7–10)	NS
ScvO_2_ at ECMO full flow (%)	69 (64–75)	67.5 (62–72)	73 (66–76.5)	NS	**73 (69–76)**	**66 (61–72)**	**0.0358**
AntiXa (UI/mL)	0.31 (0.29–0.33)	0.32 (0.29–0.33)	0.31 (0.29–0.32)	NS	0.30 (0.29–.033)	0.31 (0.29–0.32)	NS
Time on ECMO (Days)	20 (10–37)	14 (9–37)	30 (20–37)	NS	24 (10–33)	17 (10–56)	NS
Membrane changes/patient, nb (%)	2 (1–4)	1 (0–3)	2 (1–4)	NS	2 (1–3)	1 (0–4)	NS
Tracheotomy, nb (%)	22 (54)	11 (46)	11 (65)	NS	12 (71)	11 (46)	NS
CVVHDF	3 (8)	2 (8)	1 (6)	NS	0 (0)	3 (13)	NS
COVID-19 adjunctive treatment							
Hydroxychloroquine, nb (%)	10 (26)	5 (21)	5 (29)	NS	**10 (59)**	**0 (0)**	**<0.0001**
Lopinavir + Ritonavir, nb (%)	9 (22)	4 (17)	5 (29)	NS	**9 (53)**	**0 (0)**	**<0.0001**
Remdesivir, nb (%)	8 (20)	5 (21)	3 (18)	NS	3 (18)	5 (21)	NS
Dexamethasone, nb (%)	28 (68)	17 (71)	11 (65)	NS	**5 (29)**	**23 (96)**	**<0.0001**
Tociluzimab, nb (%)	22 (54)	15 (63)	7 (41)	NS	**2 (12)**	**20 (83)**	**<0.0001**
Meduri, nb (%)	27 (66)	13 (54)	14 (82)	NS	14 (82)	13 (54)	NS
COVID-19 related complications							
Pneumothorax, nb (%)	14 (34)	**5 (21)**	**9 (53)**	**0.0327**	3 (18)	11 (48)	NS
Pneumomediastinium, nb (%)	8 (20)	6 (25)	2 (12)	NS	1 (6)	7 (30)	NS
Pulmonary embolism, nb (%)	10 (26)	6 (25)	4 (24)	NS	**0 (0)**	**10 (43)**	**0.0022**
Ventilator-associated pneumonia, nb (%)	38 (93)	22 (92)	16 (94)	NS	**14 (82)**	**24 (100)**	**0.0325**
Aspergillosis, nb (%)	14 (34)	9 (38)	5 (29)	NS	3 (18)	11 (48)	NS
ICU length of stay (days)	36 (21–45)	29 (17–45)	37 (31–44)	NS	33 (21–41)	37 (21–67)	NS
Hospital length of stay (days)	45 (33–69)	45 (30–70)	42 (33–60)	NS	42 (32–56)	45 (38–74)	NS
Major adverse event, nb (%)	2 (5)	0 (0)	2 (12)	NS	2 (12)	0 (0)	NS
90-day survival, nb (%)	**24 (59)**	**24 (100)**	**0 (0)**	**<0.0001**	**6 (35)**	**17 (74)**	**0.024**

BMI: Body Mass Index, COPD: Chronic Obstructive Pulmonary Disease, IMC: Intermediate Care, ICU: Intensive Care Unit, Hb: Hemoglobin, WCC: White cell count, N/L: Neutrophil/Lymphocyte, CRP: C-Reactive Protein, Us-Troponin: Ultra-sensitive troponin, ECMO: Extracorporeal Membrane Oxygenation, HR: Heart Rate, MAP: Mean Arterial Pressure, P/F: PaO_2_/FiO_2_, Vt: Tidal Volume, PBW: Predicted Body Weight, PEEP: Positive End-Expiratory Pressure, iNO: Inhaled Nitric Oxide, VA: Veno-Arterial, VV: Veno-venous, ScvO_2_: Central Venous Oxygen saturation, CVVHDF: Continuous Veno-Venous Hemodiaifltration, nb: number, SAPS II: Simplified Acute Physiology Score. Bold: a statically significance.

**Table 2 jcm-12-00230-t002:** Association between days under mechanical ventilation and total days of ventilation before ECMO and 90-day mortality.

	90-Day Mortality in Patients under ECMO OR (95% CI)	*p*
**Time under MV before ECMO**		
≤3 days	Ref.	0.03
>3 days	4.9 (1.1–20.9)	
**Time under NIV and MV before ECMO**		
≤7 days	Ref.	
>7 days	4 (1.1–15.7)	0.049
**SAPSII**	1.03 (0.9–1.1)	0.3
**Plateau pressure**		
<28 cmH_2_O	Ref.	
≥28 cmH_2_O	1.1 (0.2–6)	0.9

CI: Confident Intervalle; MV: mechanical ventilation; ECMO: Extracorporeal Membrane Oxygenation; NIV: non-invasive ventilation; OR: odds ratio.

**Table 3 jcm-12-00230-t003:** Association between having an ECMO team in charge of the patient and 90-day mortality.

	90-Day Mortality OR (95% CI)	*p*
ECMO team in charge	0.2 (0.001–0.2)	**<0.01**
SAPSII	1 (0.9–1.1)	0.9
Duration of MV before implantation of ECMO > 72 h	1.3 (0.2–9.2)	0.8

SAPS II: Simplified Acute Physiology Score. Bold: a statically significance.

## Data Availability

The datasets used and/or analyzed during the current study are available from the corresponding author on reasonable request.

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
