# Peer review of "A Dedicated Expert ECMO-Team and Strict Patient Selection Improve Survival of Patients with Severe SARS-CoV-2 ARDS Supported by VV-ECMO"

_jcm, 2022, doi:10.3390/jcm12010230_

Round 1

Reviewer 1 Report

The presented study describes the positive effect of early VV- ECMO in COVID-19 patients with hypoxemia and the benefit of patient care by experienced personnel. Procedure timing is indeed always an important factor as to turn around the patient’s conditions prior to the point of no-return.

Having experienced staff is always the optimal situation and is definitely worth being emphasized over and over again. The title is maybe a bit misleading as aside from timing there is not really mentioned what the strict patient selection contained.

While the discussion heavily focusses on ECMO volume/cases, the timing is no longer discussed. Overall the message is a bit blurry of this manuscript and should be clarified.

Furthermore, looking at all the parameters in which patients in the first and in the subsequent waves differ, it might be a bit quick to only associate timing and personnel with the improved outcome. Patients in the second wave and afterwards appeared to present with a much better pre-ECMO condition (less CRP, lower Troponin levels, lower D-Dimers) than patients in the first wave, which may mainly be due to the adapted medical management of COVID-19 patients and changes in ventilation strategies upon hospitalization.

Vaccination status of the patients is not mentioned, as the study included patients all the way to 2022, this might be worth adding.

Overall the paper does not read very well and needs major re-writing.

Abstract

Gender distribution not mentioned, Vaccination status post 2021 not mentioned.

Qualified team in the ECMO Procedure: does that refer to placing the ECMO (choice of vascular access), Monitoring the ECMO (physiologic understanding) or to the timing of placing the ECMO based on the patient’s condition?

Introduction

The introduction is rather short, please provide more information on what was already found to have changed over time that improved survival in ECMO patients (aside from the personnel’s expertise): severity of respiratory illness decreased with different strains/vaccination? Adaptation of ventilation protocols prior to ECMO?

P2 line 51, please specify that those were all COVID19 positive patients

Material and Methods

P2 line 65,66: This sentence is unclear. The mobile ECMO team was responsible for the patients in Geneva, other hospitals which lacked the ECMO expertise plus they transported patients from other hospital to Geneva? Please rewrite for better understanding.

P2 line 66: Please give a number of total patients included.

P2 line 68,69: This is contradictory to what you mention in the abstract where you state that only VV-ECMO patients were included. Here you state VA-ECMO patients were included as well. Please clarify, as the type of ECMO could further cause a difference in patient survival rate.

P2 line 87: Is latin canton an official term? Otherwise I would refer to French and Italian-speaking regions

P2 line 88-91: I would rephrase this sentence to “Although a national coordination in intensive care medicine was established quickly to identify the number of ICU beds across the country,  it was only during the second wave that this coordination unit was commissioned to help transfer patients from overburdened centers to others with more beds available.”

P2 line 89: I would remove the term precisely as the number of ICU beds fluctuated not only due to increasingly postponing elective surgeries (or decreasing this measure in the following waves), but also due to shortness of staff able to care for the ICU bed.

P2 line 91: please change the term saturated to overburdened.

P2 line 93: please change to ECMO specialists of various centers developed and distributed common guidelines, in particular to non-ECMO centers…

Is there a reference to these guidelines you can add?

P3 line 99: To the 10 nurses refer to your ICU in Geneva? Please clarify when you talk about your hospital and when you talk about national measures. Was this during the first wave or in light of future waves? This entire paragraph appears a bit confusing. Please consider re-writing.

P3 line 101: with expertise in managing patients on ECMO or does this refer to critical patients on ventilators as well?

P3 line 105: What does “in cells” mean?

P3 line 114: from the second wave on

P3 line 116: I assume the “fulfilling the criteria” refers to the patient, the way it’s phrased it may also refer to the non-ECMO center. In this case these criteria would have to mentioned somewhere.

P3 line 117: The ECMO consultant is one of the two intensivists physicians?

P3 line 125: where those the recommendation of MV during ECMO or MV in general?

P4 line 188: all patients according to the BMI were overweight or obese, or am I wrong?

P4 line 189, please introduce the abbreviation “IMC” here

P5 line 207: The septic shock does not fit in this sentence, as it reads as “with no possibility of withdrawal from septic shock”, please rephrase.

P5 line 209-219: please list the values for survivors non-survivors always in the same order

P5 line 218: What does the value of following or monitoring of the patient represent? (18 vs 5), is there a unit?

Table 1: The average age is listed as 57 (52-63), however under First wave it then states 80 (70-88) and other waves (86 (85-92). What does this age during first wave and other wave mean? This is confusing.

Obesity: The numbers of the first wave and the other waves equals to 16, in the same line only 14 patients are listed as presenting with obesity.

Asthma/COPD: the number of patients don’t add up. Total 3: survivor 1, non survivor 1 (=2)

Time from symptoms to Hospital Admission (Days): maybe put here mean (range). It is interesting that in the following wave the time from symptoms to hospitalization was increased, what was the reason behind this you believe?

Why did patients show a highr Hg concentration pre-ECMO in the other waves but not the first? Higher fluid infuisons?

What does the “ECMO TEAM in charge, nb (%)” describe?

Why were large cannulas chosen for the other waves in comparison to the first wave?

Why were almost half the patient tracheotomized and the other half not ?

P8 line 245: the word comparing is redundant please delete

P8 line 246: maybe replace and with nonetheless, the survival rate was higher…

P9 line 265: missing the word “patients” (COVID-19 patients)

Discussion

P9 line 268: What do you mean by we limited the implantation to patients who had MV for less than 7 days? Was the standard not adapted to put patients on ECMO before 7 days of MV or what was the criteria to continue to MV patients without putting them on ECMO?

P9 line 275: What do you mean by “the place”? The role? The benefit? Can you please reference this statement if it is a debated topic

P10 line 287: please replace reinforced with a term like “confirmed”, “underlined”, “emphasized”

P10 line 288: I am not sure what this means “indications were provided” the ECMO team defined the criteria for a patient to be put on ECMO? Isn’t the part of the same criteria not redundant?

P 10 line 290: was not associated (please delete independently it is contradictory). Did you mean it was center-dependent associated with survival? This sentence is not clear, you want to say that number of ECMOs is associated with better outcomes only on a single center level? Or do you mean the volume of particularly VV-ECMO is associated with a better outcome but not any of the other type of ECMOs (VA. VAV). Then you would also have to specify this on line 294 to emphasize the importance of having expertise in VV-ECMO. Please try to rephrase this paragraph.

P 10 line 295: Maybe you can give an idea on what the limitations are (how much more staff would you need to have 100% of patients on ECMO being cared for by highly experienced personnel. Is there a potential need for a new educational branch in nursing?)

Author Response

Revised version R1-jcm-2049488: A dedicated expert team and strict patient selection improve survival of patients with severe SARS-CoV-2 ARDS supported by VV-ECMO”

We thank you for the careful review of our manuscript and for the constructive criticisms. The suggestions were very helpful and we appreciated them. We have revised the manuscript as suggested.

You will find in attached files our point-by-point responses to your comments. We also include a revised version of our manuscript, containing a document with changes highlighted in red and a clean version with all changes accepted.

All authors have contributed significantly to the present work. They have read and approved submission of the manuscript,

We thank you by advance for reconsidering our manuscript and hope that you will now find it suitable for publication in the Journal of Clinical Medicine. Additionally, we thank you for your generous consideration.

Looking forward to hearing from you.

Raphael Giraud, MD, PhD

Reviewer 2 Report

The authors conducted a retrospective study in one health facility and report that favorable patient outcomes occurred under certain conditions. In order to appreciate the findings of the study, the authors must clarify several issues as outlined below.

Title: What is meant by a) a dedicated team b) strict patient selection is not clear in the body of the manuscript.

Line 27: What groups do the lengths of stay refer to, survivors and non-survivors?

Line 32: A lot of emphasis is given to the role of healthcare workers’ expertise/experience. However, are there alternative explanations for the apparent improved survival? This is an important question noting that the treatment regimen for the initial wave was different from that during the subsequent waves.

Line 61: The methods section is not clear particularly with regards to expertise of healthcare workers administering ECMO as well as those managing the patients. Specifically, it is nor clear how many patients received specialized care and how many did not receive such care?

Line 187-188: How are the percentages reported here derived? In other words, what is the denominator?

Line 188: What is the source of the data on time from symptom presentation to hospital admission? This is an important question especially given that informed consent was waved for this study?

Line 200: Why switch to mean from median? What are the values in brackets?

Line 215: This is being repeated. There are several instances of such repetitions in the manuscript. Correct unless the redundancies are justified.

Line 218: Could the difference in incidences of pneumothorax as well as others not related to ECMO and health care work experience explain the differences in survival?

Table 1: It is quite helpful to note that a part from a few cases, survivors and non-survivors appear to not differ in several characteristics.

Line 242: What does more responsibility mean? I am a bit concerned about a claim or evidence such as this one because it raises serious ethical concerns.

Line 262: This is not clear at all: What is 90-day survival and what is a true 90-day survival? The present study has reported 90-day survival of 59%, for those who survived for 90 days, which was the census day in this study. Is there a different 90 day cut-off?

Line 273: It would be great for the authors to account for the improved odds of survival when ECMO is initiated early. In others words, mechanistically speaking, what is the explanation for improved survival when ECMO is initiated early?

Line 294: What was the distribution of patients who received expert ECMO and those who did not receive ECMO?

Line 316: Please note that the component of patient selection is missing from the conclusion yet it features in the title.

Line 332: I am quite surprised that a study that used data collected by others and one that is based on information on a pandemic has no one or institution to thank.

Author Response

(The authors gave the same response as above.)
